# Assessment of the Impact of Lean Tools on the Safety of the Shoemaking Industry

José Carlos Sá [1,*], Leonardo Soares [1], José Dinis-Carvalho [2], Francisco J. G. Silva [1] and Gilberto Santos [3]

[1] ISEP, Polytechnic of Porto, 4200-072 Porto, Portugal; 1180061@isep.ipp.pt (L.S.); fgs@isep.ipp.pt (F.J.G.S.)
[2] Production and Systems Department, University of Minho, Campus de Azurém, 4800-058 Guimarães, Portugal; dinis@dps.uminho.pt
[3] School of Design, Polytechnic Institute Cávado and Ave, Campus do IPCA, 4750-810 Barcelos, Portugal; gsantos@ipca.pt
* Correspondence: cvs@isep.ipp.pt

**Abstract:** Both the Lean philosophy and occupational safety and health have been widely studied, although this has usually been carried out independently. However, the correlation between Lean and occupational safety and health in the industrial context is still underexplored. Indeed, Lean tools can be applied to ensure the best safety environment for workers in each kind of manufacturing process, and this deserves to be studied. The study described here aims to understand the influence of each of a set of four Lean tools used in an industrial context with a strong manual labor component, seeking to determine the influence of each of these Lean tools on the increase in safety obtained through their application. For this purpose, four Lean tools that are quite commonly applied are selected, taking into account previously presented work that pointed to the positive influence of the application of each of these tools on worker safety: total productive maintenance system, Gemba walk, visual management and Yokoten. This study aims to apply these Lean tools and to analyze their impact on productivity, and then, on the safety of a company selected as a target in order to validate the concept. For this purpose, a new tool is created. In the first instance, the tool analyzes the current state of the productive process and the safety level through the study of the risk levels detected in the plant. In terms of productivity results, a reduction between 7% and 12% in cycle time is achieved in four areas of the plant. The feedback from employees showed increased satisfaction with the processes' simplification. To conclude, a 50% reduction in the number of work accidents per month is observed as a result of the implementation of Lean tools. The influence of the selected Lean tools on increasing both productivity and safety is clear, and our results prove the selection of tools to be largely adequate.

**Keywords:** Gemba walk; Lean safety; orthopedic footwear; total productive maintenance; value stream mapping; Yokoten; overall equipment effectiveness

## 1. Introduction

Lean manufacturing is highly used in industrial and service sectors today in order to eliminate waste (*muda*) and normalize processes [1]. Through a number of tools, the Lean philosophy has gained followers all over the world due to its effectiveness in several areas, with particular attention paid to the flexibility brought to production systems [2–4] and maintenance [5–7]. In addition to their negative impact on people's well-being and morale, injuries at work can in some ways be considered waste because they negatively affect productivity [8,9]. However, it can be stated that occupational safety and Lean philosophy are perfectly interconnected and aligned [10]. It is well known that the use of hand tools is connected with the appearance of hand injuries, namely, musculoskeletal problems in the upper body [11], and Lean tools try to avoid those situations, creating better work behavior. At the safety level, it is important to share information regarding occupational diseases



outbreaks and accidents that occur, with the aim of preventing their recurrence for the same reasons [12]. Workplace safety has been gaining prominence in the industrial environment, changing the way safety management systems are managed [13]. Due to the nature of their job functions, there are more chances of a work accident happening when workers are left to make decisions regarding their safety [14]. Work-related musculoskeletal disorders are disorders in muscles and tendons that result from working conditions. They occur due to multiple factors, such as the physical load applied or by psychological factors [15].

Next, a review of the literature that served as the basis for this work is presented in order to provide the reader with some of the research previously published on this topic. The review is divided into two main topics: safety and Lean tools.

### 1.1. Occupational Safety and Health

Several authors have addressed security problems by integrating Lean tools. This integration has given rise to new tools that result from frameworks developed based on different approaches and a combination of Lean tools. Marques Filho [16] created a new tool called SVSM (safety value stream mapping), according to which the author allows the reinforcement and removal of the best of two tools, namely VSM and SSM (safety stream mapping). This SVSM tool allowed the representation and identification of traditional waste and existing occupational safety problems. To validate the success of this tool, the author stated that it needed to be tested in different industrial sectors. Brito et al. [17] developed an operational tool to help researchers and professionals prioritize and evaluate ergonomic and safety implementations, as well as overall conditions, in an integrated way. The authors found that the highest scores obtained were the result of good interactions between the Lean system, ergonomics and safety. Aqlan et al. [18] integrated the principles of Lean manufacturing and ergonomics in order to redesign and improve the internal transport process of electronic parts in a company. Based on the concepts of Lean philosophy and ergonomics, the transport flow was redesigned. The new design introduced the "Lean tool" poka-yoke, which prevents operators from stacking cards, thus eliminating safety and ergonomic risks. Pereira et al. [19] developed a project whose main objective was to improve operators' working conditions and define the most suitable augmented reality (AR) for each material handling and movement process. To this end, they developed a methodology called risk assessment for ergonomics and safety in logistics (RAES-Log) to analyze and define the requirements for implementing AR so as to mitigate existing risks and improve ergonomic conditions. Although aware of the difficulties they faced in implementing the Lean philosophy, the company's ergonomists recognized the importance of this philosophy in improving working conditions in the logistics area. For Crema et al. [20], the reduction in resources used in healthcare and the need to ensure high levels of quality has led hospitals to develop projects that report multiple performances. In order to try to improve patient safety and efficiency at the same time. Thus, "Lean & safety" projects (L&S projects) can be implemented, combining Lean healthcare management (HLM) and clinical risk management (CRM). In the case study presented by the authors in the article, through the Lean healthcare management project, actions were implemented that led to efficiency, eliminated costly and unnecessary activities and materials, but above all led to positive results in terms of patient safety. Near misses and errors were completely reduced by 84.38%, enhancing patient safety. Pereira and Xavier [21] developed the SMILE approach in the workplace and developed a new conceptual model to be implemented in organizations to achieve a safety culture. According to the authors, adopting this integrated Lean safety model in an organization can definitely help the company to reap good benefits in terms of an accident-free organization, performance excellence, and huge financial benefits. The project team (Six Sigma team) should be trained to solve the safety-related problems in industry using the DMAIC (Define – Measure – Analyse – Improve – Control) approach. Ateekh-ur-Rehman [22] presented a study that aimed to use the Six Sigma philosophy to identify and reduce the occurrence of accidents at work in a company. According to the author, in order to achieve zero injuries or minimize workplace accident rates

and/or financial losses, it is necessary to use control methods. This study demonstrated the effectiveness of the DMAIC methodology approach of the Six Sigma philosophy in reducing occupational safety risks. In the article presented by Tsung and So [23], DMAIC from the Six Sigma philosophy was used to analyze historical data on injury rates in a company. The authors chose to focus the project on "manual handling injuries", as these accounted for almost 50% of lost time injuries (LTIs) and medical time injury (MTIs). The critical factors were then identified, as well as the most dangerous activities related to workplace injuries. Using the continuous improvement procedures outlined in the Six Sigma philosophy, they identified the critical factors and defined action plans that can be used to mitigate the safety risks. Umar et al. [24] took a very interesting approach to analyzing the impact of lean tools on safety. They developed a framework that provided additional insight into the 3M concept of lean with the integration of ergonomics. The research carried out by the authors showed that the three concepts underpinning the 3Ms, which are muda (waste), mura (inconsistency), and muri (over-burdening), can also be seen from the worker's perspective in a way that considers the physical and mental resources of the workers. These are vital to the performance of the workers and will directly influence the overall performance of the processes in the organization. Other authors have also referred in a more general way to the harmful aspects of problems related to occupational safety and health. Abu Aisheh [25] argues that work accidents can be seen as waste, being obstacles to a smooth process. By making the process more efficient, with reduced cycle and material handling times, it reduces workers' exposure to sources of possible accidents [26].

For Moldovan [27], Lean and Six Sigma methods can be used to improve labor practices to, in the end, improve medical organizations, which will benefit the safety of patients.

Singh [8] concluded in his study that, with the implementation of Lean tools, in particular the 5S and poka-yoke, it is possible to improve the safety and health of employees. Mutaza [28] concluded after the literature review that, theoretically, the 5S tool helps in reducing occupational accidents, but that the tool however should be accompanied by systematic monitoring, such as internal audits. Moreover, James [29] observed the reduction in risk in two areas of modular house construction. Thus, it was concluded that it is a good tool to improve productivity and safety.

### 1.2. Lean Tools with Interest for the Present Work

#### 1.2.1. Value Steam Mapping (VSM)

Visual stream mapping is very useful for visualizing and quantifying the complex production process on the shop floor [30]. In addition to enabling the visualization of the production process, it also allows the visualization of cycle times, inventory buffers, and information flow, as well as the transformation of raw material into finished products [31,32]. Olakotan [33], after applying the VSM tool to the clinical decision support system that generates medication alerts, detected that it was possible to identify several wastes throughout the clinical procedures. Arifin [34], in an attempt to identify the error factors in laboratories, concluded that Lean tools are applicable to assessments of the laboratory process in a structured and rigorous manner.

#### 1.2.2. Visual Management

Visual management is a tool for decision support and process improvement [35]. This tool leads to an improvement in the efficiency and effectiveness of communication, displaying indicators [36]. The impact of visual management in safety was also approached by Babur [12] and Sá [37]. The first author [12] created a roadmap in a Turkish shipyard using Lean tools to significantly reduce work accidents, absenteeism, and manufacturing costs. The second one [38] applied two tools, visual management and 5S, allowing for a better organization of the process, a 40% reduction in activities considered wasteful, and productivity levels of 74% and 87% to be reached in the finishing and cabinetry sections, respectively.

### 1.2.3. Yokoten

From the safety point of view, Yokoten enabled the dissemination of measures aimed at preventing and, in this sense, reducing absenteeism, the development of occupational diseases, and the occurrence of accidents at work [12]. In his case study, Machikita [38] concluded that 5S and Yokoten can make organizational routines necessary to improve knowledge transfer capabilities. In a case study performed by Fernando [39] interviewing a company, the company itself stated that Yokoten focuses a great deal on sharing in order to identify waste and systematically reduce it.

### 1.2.4. TPM

TPM (Total Productive Maintenance) idealizes a scenario of zero stoppages, zero defects and zero accidents. This requires a commitment and culture of continuous improvement from operators to the top management [40]. Biazzo [41] concluded in his study that the application of tools such as 5S, TPM and Kaizen has enabled the mitigation of multiple risks and hazards, and the elimination of wasteful activities. Sá [42] also reported significant gains in terms of OEE (pverall equipment effectiveness) and the availability of equipment in two different sectors of a metalworking industry after the implementation of just some pillars of the TPM principles.

### 1.2.5. Gemba Walks

The Gemba walk allows managers to go out into the field and identify and try to understand the main challenges and problems facing the shop floor. It is hoped that with this learning process, where problems are analyzed and solutions are created, the outcomes will result in improved operational performance and capacity [43].

### 1.2.6. Standard Work

Standard work is a tool that aims to reduce waste and maximize individual and team performance through the creation of procedures [44]. For MÍkva [45], procedures allow workers to be informed about the best method of carrying out the task. Each change to the process will only be complete when creating or updating these. The use of procedures allows companies to reduce error variation, increase security, improve communication and increase the visibility of problems [46].

Considering the sectors analyzed in the reviewed works and the fact that no studies were found that specifically dealt with the footwear industry, there is a gap that this work intends to fill. In fact, the footwear industry still involves a high amount of manual work, with obvious risks to workers given the small size of the product, the proximity of risk factors to workers' limbs, and fatigue from routine operations repeated countless times during each work shift. Taking this scenario into account, it was considered appropriate to extend some previous studies to this sector and integrate new concepts more adapted to the specific characteristics of this type of industry.

The main goal of this work is to study the impact of the application of precious selected Lean tools to improve productivity and occupational safety. To validate the research, an orthopedic footwear company was selected, because, traditionally, this kind of companies present high level of human labor, some competitiveness problems, and some lacks in safety procedures. Thus, the work aimed to analyze the production process and the safety level of the company, apply the Lean tools from the set of tools described in the previous section that better fit the problems identified, measure the results, and draw conclusions about the application of the tools.

## 2. Materials and Methods

In this research, the methodology known as action research was applied because it is intended to solve an existing problem and extrapolate the corresponding knowledge to future similar situations. Mello et al. [47] state that "research-action is the production of knowledge guided by practice, with the modification of a given reality occurring as part of the research

process". Following the procedures used by Marinho et al. [48] Mourato et al. [49], and Martins [50], the methodology is composed of five phases, as can be seen in Table 1.

**Table 1.** Presentation of the stages of the methodology action research.

| Step | Content |
|---|---|
| Diagnosis | An SVSM map was built to be able to have an overview of productivity and safety on the shop floor by showing the cycle time, the changeover time, and the risk of each area. |
| Action planning | Based on the literature, Lean tools Gemba walk, TPM, visual management and Yokoten were chosen, and their implementation was planned. |
| Implementation | The Lean tools were implemented, and a summary of their actions was presented. |
| Evaluation | Taking into account the indicators used in the diagnosis, results after the Lean tool implementation were compared to those initially obtained. |
| Monitoring | Definition of indicators to control and monitor the situation in the future were created. |

### 2.1. Diagnosis

In order to diagnose the initial situation and performed the first stage of the action research methodology, it was necessary to understand the influence that each productive section had on the workflow in terms of productivity and safety. As a first approach, we measured the cycle time (C/T), the changeover time (C/O) and the distance travelled (D/T) of each area. As a second approach, a risk map was built for each area. In Table 2, it is possible to observe the production parameters mentioned above for the different areas. Data were collected through the MES (manufacturing enterprise system), such as productive data, between the initial and final dates for analysis, which will be provided later in this study. The accidents at work are also reported in a specific module of the MES used by the company. These have undergone some customization in terms of the specific requirements of the company. The same procedure was used to perform analysis after the Lean tool implementation.

**Table 2.** Distribution of the number of workers, cycle time, changeover time and distance travelled for five production areas of the company used as target for this work.

| Areas | Number of Workers | C/T (min) | C/O (min) | D/T (m) |
|---|---|---|---|---|
| Translation & Supply | 8 | 48.30 | 3 | 72 |
| Lasts | 5 | 31.00 | 5 | 32 |
| Modeling | 9 | 125.3 | 8 | 10 |
| Cut & Stitching | 26 | 197.3 | 15 | 32 |
| Assembly & Soles | 20 | 133.3 | 12 | 38 |

The method used to analyze the risks for each task was the simplified MARAT (methodology for risk assessment and accidents at work) [51]. The classification of hazards is based on the calculation of the risk level (NR). The calculation is performed on the basis of the relationship between variables that assess probability, consequence, exposure, and disability. The risks of each area were evaluated in five different categories: mechanical (NR MEC), physical (NR FIS), ergonomic (NR ERG), chemical (NR QUIM) and biological risk (NR BIO). Table 3 summarizes the average risks by category and by production area, showing the total average risk in the last column.

**Table 3.** Distribution of average risk levels divided into five categories regarding six stages of the production process.

| Areas | NR BIO | NR ERG | NR FIS | NR MEC | NR QUIM | Mean |
|---|---|---|---|---|---|---|
| Conveying & Supply | 0 | 94 | 150 | 22 | 20 | 106 |
| Lasts | 0 | 75 | 0 | 91 | 84 | 83 |
| Modeling | 25 | 200 | 0 | 30 | 0 | 85 |
| Cut & Stitching | 0 | 110 | 0 | 163.5 | 65 | 123 |
| Assembly & Soles | 0 | 208 | 0 | 142.5 | 60 | 137 |
| Finishing & Dispatch | 0 | 82,5 | 0 | 60 | 50 | 61 |

Once all the information was gathered, the SVSM was built, which can be seen in Figure 1.

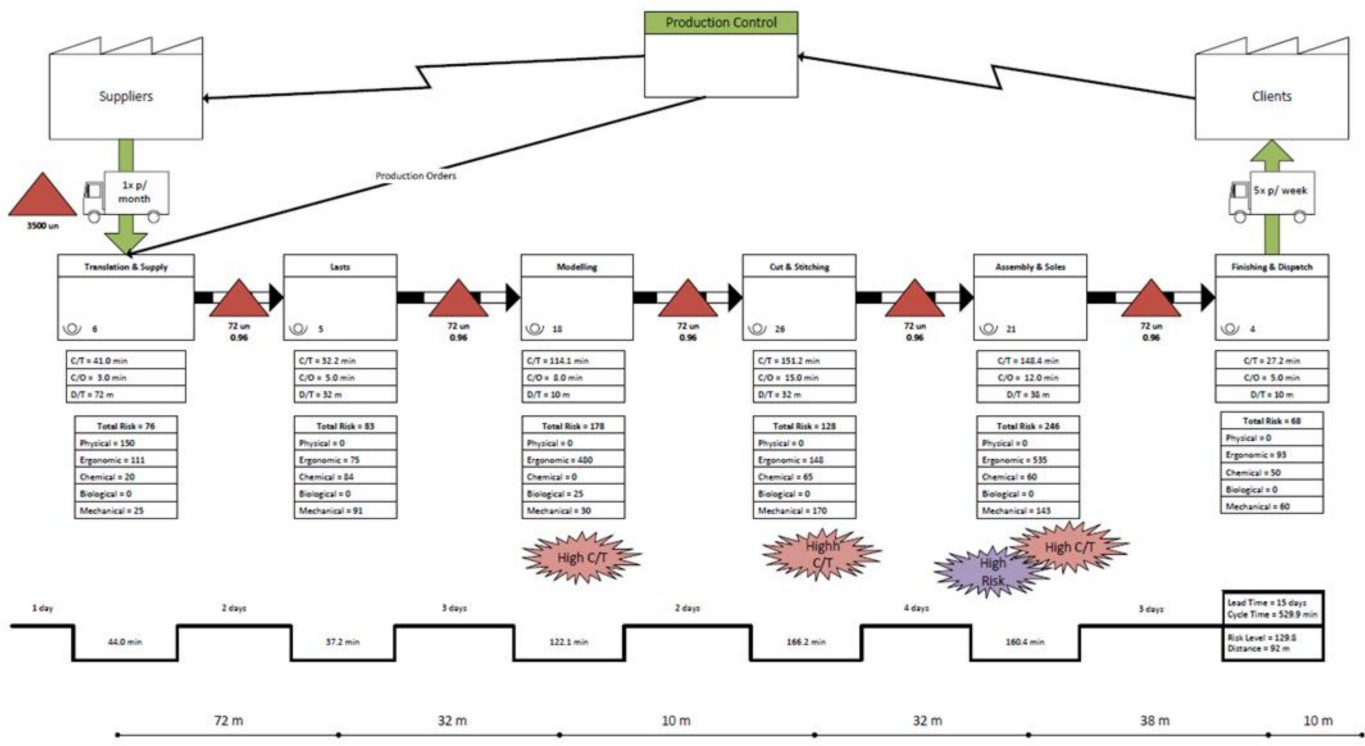

**Figure 1.** SVSM map before tool implementation.

### 2.2. Action Planning

It was hypothesized whether Lean tools would have a positive impact on the company's productivity and production flow and, consequently, what effect they would have on safety. For this study, based on previous approaches referred to in the literature, four Lean tools were chosen: TPM, Gemba walk, visual management and Yokoten.

The implementation of a TPM system aims to reduce the waiting time for a problem to be solved and to contribute to the better organization of maintenance operations in order to manage them by priority, duration, or deadlines. The Gemba walk aims to identify and reduce all types of production waste, as well as identify and act on possible safety risks. Visual management aims to speed up the movement of late products through a productive area so that, at the end of the process, companies meet the deadlines agreed with the customer. Finally, Yokoten intends to act on safety in order to reduce the incidence of work accidents. It is believed that these four tools together will be enough to face the usual problems of productivity and safety observed on the company's shopfloor and that this paper intends to identify and help to overcome.

After the implementation of Lean tools, the following positive results are expected: a reduction in the cycle times of each area with the application of Gemba walk and visual



management Lean tools and a reduction in the risk levels in each area with the implementation of TPM and Yokoten Lean tools. It must remain clear that the only component that can be reduced in a work risk is the probability of it happening (the consequence remains unchanged).

### 2.3. Lean Tool Implementation

Next, the way in which each Lean tool was implemented in practice is described, one by one, in the specific case of the company selected to implement this study.

### 2.3.1. TPM

To highlight TPM orders, it was decided to create identifying labels, as shown in Figure 2. The three categories are the following: Maintenance (red), Safety (green) and Production (blue). About 264 tags were opened, distributed as follows: 107 for Maintenance, 95 for Production and 62 for Safety. Figure 3 presents the distribution of open tags by problem location.

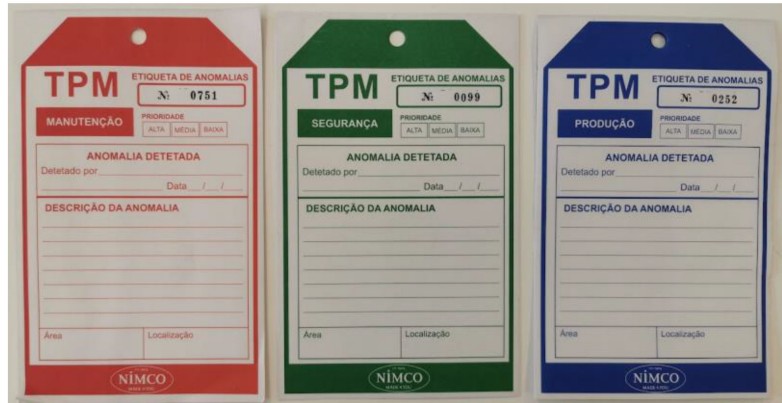

**Figure 2.** Template of TPM labels for the categories Maintenance, Safety and Production, respectively.

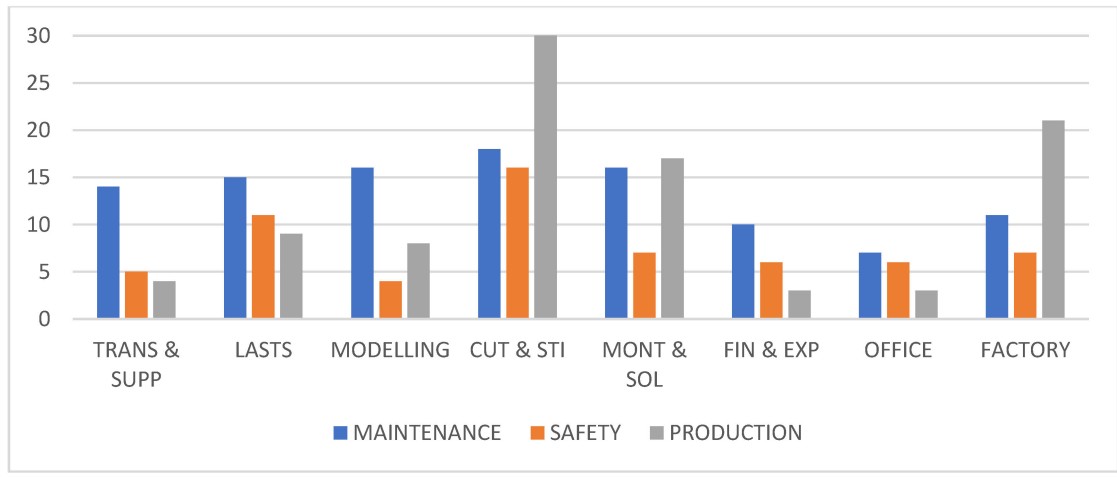

**Figure 3.** Distribution of open tags by problem location.

### 2.3.2. Gemba Walk

The second Lean tool implemented was the Gemba walk. For the implementation of this tool, a specific team was created. Moreover, the duration of the experience was set at three days, during which the team made a distribution of subjects to be given greater attention, in addition to a general follow-up with workers in this area and their tasks. In terms of deadlines for solving each task, this was determined considering the availability of those responsible, the priority of the task and its complexity. The areas chosen were

the Supply, Conveying (Trans), Lasts, and Finishing. The graphs in Figures 4 and 5 show, respectively, the distribution of responsibility for resolving all the reported problems by each area visited and the distribution of deadlines for completion.

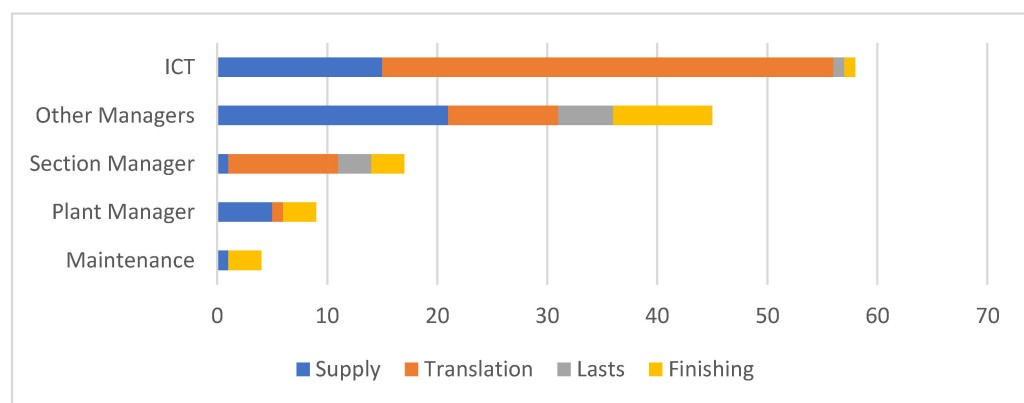

**Figure 4.** Distribution of responsibility for resolving all the identified problems.

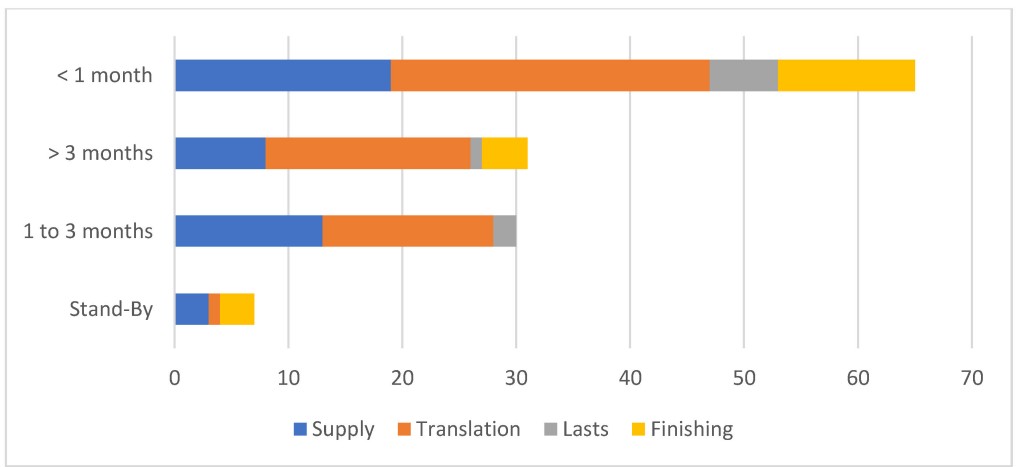

**Figure 5.** Distribution of deadlines for conclusion.

### 2.3.3. Visual Management

Regarding the application of the visual management tool, the study assessed the presence of late products in the areas because they are not a permanent focus of attention. It was decided to create a card printed on brightly colored paper indicating that a specific pair is a late pair and needs to be completed urgently. In an experimental phase, the Soles area was chosen (Figure 6).

### 2.3.4. Yokoten

Regarding safety issues, it was decided to create a document to inform the company workers about accidents and near-accidents at work. The form adopted was a table containing the following information: the description, the material agent of the activity and the type of consequence caused to the worker in terms of injury and days off.

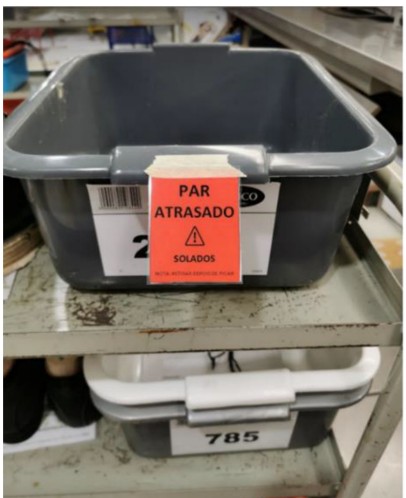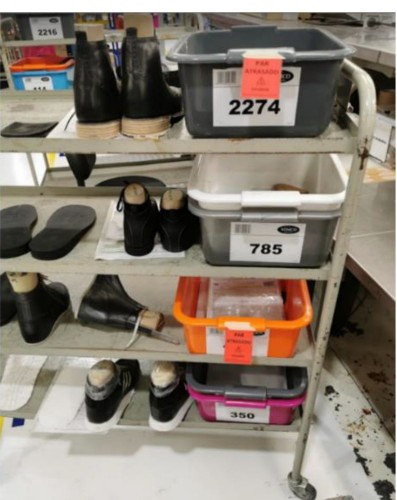

**Figure 6.** Application of the Visual Management as solution for late pairs on the shop floor (Par Atrasado = Late Pair).

## 3. Results

This section depicts the "Evaluation" phase of the methodology followed. Table 4 presents the values for the mentioned parameters before and after the implementation of the mentioned tools.

**Table 4.** Comparison of the values recorded for cycle times, changeover time and distance travelled before and after the implementation of the tools.

| Parameter | Time Frame | Translation & Supply Chain | Lasts | Modeling | Cut & Stitching | Assembly & Soles | Finishing & Dispatch |
|---|---|---|---|---|---|---|---|
| Workers | | 6 | 5 | 18 | 26 | 22 | 4 |
| | | 6 | 5 | 16 | 26 | 21 | 4 |
| Dif | | 0 | 0 | −2 | 0 | −1 | 0 |
| C/T (min) | | 41.0 | 32.2 | 114.1 | 151.2 | 148.4 | 27.2 |
| | | 37.0 | 28.3 | 105.5 | 157.5 | 155.4 | 28.3 |
| Dif (%) | | −9.7 | −12.0 | −7.5 | 4.2 | 4.7 | 4.3 |
| C/O (min) | | 3 | 5 | 8 | 15 | 12 | 5 |
| | | 3 | 5 | 8 | 15 | 12 | 5 |
| Dif (%) | | 0 | 0 | 0 | 0 | 0 | 0 |
| D/T (m) | | 72 | 32 | 10 | 32 | 38 | 10 |
| | | 72 | 32 | 10 | 32 | 38 | 10 |
| Dif (%) | | 0 | 0 | 0 | 0 | 0 | 0 |

Regarding safety, first the risks present in each area were reassessed. In relation to the list drawn up at the first moment, there was only one change. The risk levels for each area were recalculated and can be seen in Table 5. The construction of the SVSM map after the application of Lean tools is presented in Figure 7.

**Table 5.** Distribution of average risk levels divided into five categories present in the production process after the application of the selected Lean tools.

| Areas | NR BIO | NR ERG | NR FIS | NR MEC | NR QUIM | Mean |
|---|---|---|---|---|---|---|
| Conveying & Supply | 0 | 109 | 150 | 22 | 20 | 75 |
| Lasts | 0 | 75 | 0 | 91 | 84 | 83 |
| Modeling | 25 | 480 | 0 | 30 | 0 | 178 |
| Cut & Stitching | 0 | 148 | 0 | 170 | 65 | 128 |
| Assembly & Soles | 0 | 535 | 0 | 143 | 60 | 246 |
| Finishing & Dispatch | 0 | 93 | 0 | 60 | 50 | 68 |

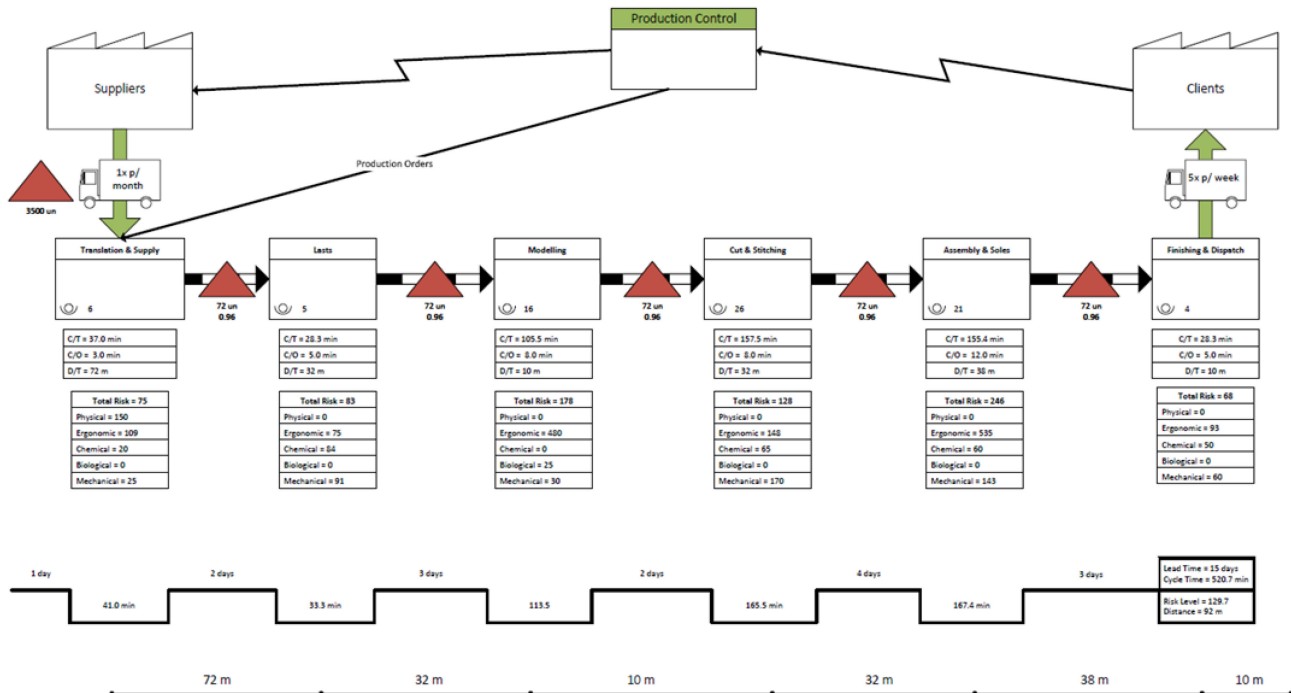

**Figure 7.** SVSM after the implementation of tools.

A study was carried out for the evolution of the number of accidents in 2021 and 2022, with a special focus on the period from September 2021 to July 2022, the period in which this project was carried out. Figure 8 presents a graph displaying the evolution of the number of accidents at work from January 2021 to July 2022. In blue represents the time before the implementation of the tools and with orange during the implementation.

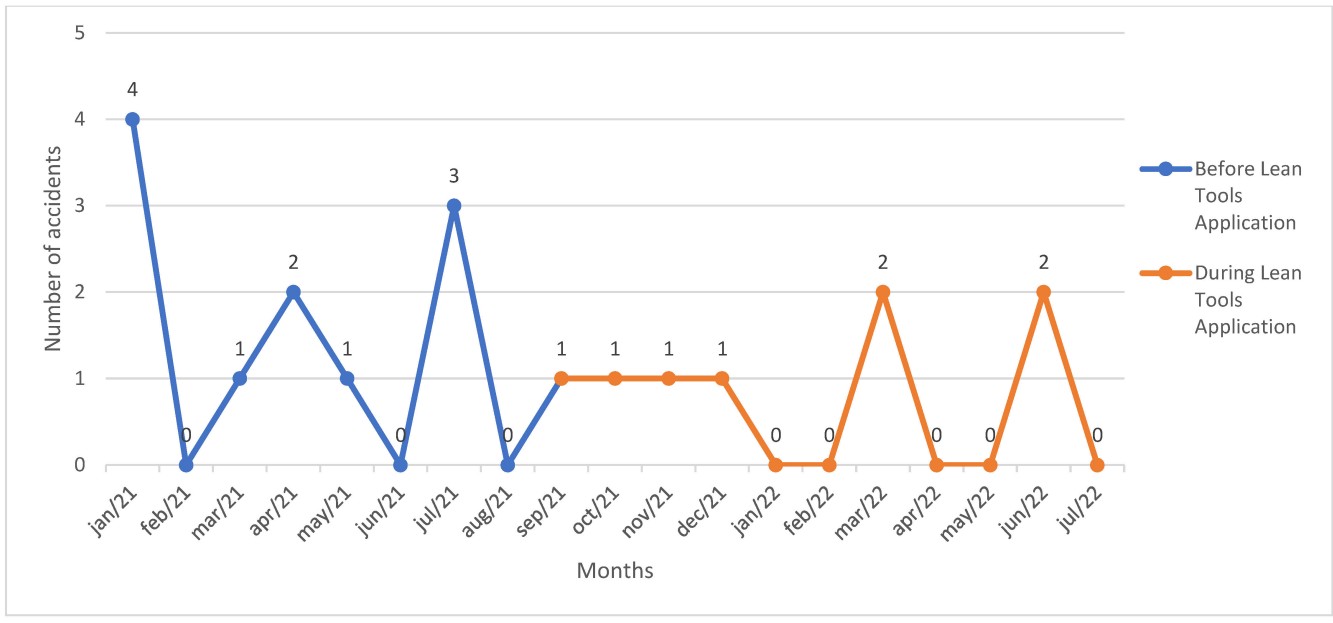

**Figure 8.** Evolution of the number of accidents at work from January 2021 to July 2022. Blue color: before implementation; Orange color: after implementation.

## 4. Discussion

The implementation of a set of Lean tools in a company dedicated to the personalized manufacture of shoes to order and still performing a high degree of manual work is a

challenge. Given that the company used to validate the concept has MES software and computerized data collection, possible deviations were minimized as much as possible. Possible errors could be related to forgetting to open or close tasks, joining different tasks in the same record, or stopping the equipment without announcing its stop (micro-stops). It was estimated that the error associated with these factors, in the overall calculation of the study, does not exceed 2%.

Analyzing the new SVSM drawn after the Lean tool implementation, the most time-consuming processes were found to be Cut & Stitching, followed by Assembly & Soles, and finally Modeling. These are areas with a very long cycle time, implying that a greater number of people are required so that the planned level of shoe production for the day can be fulfilled. It is possible to observe that the areas with a higher average risk are Assembly & Soles. The most affected area has a very high ergonomic risk due to the use of hand tools, use of machines and inadequate posture.

Analyzing Table 4, it can be seen that the first three areas underwent a positive change regarding the cycle time, thus managing to complete their work in less time. On the other hand, the remaining areas suffered a slight increase, between 4% and 5%. Gemba walk proved to be a very effective tool in terms of cycle time reduction, promoting a decrease in the cycle time in three of the four intervened areas. With regard to the Gemba walk, at first sight, the number of improvement opportunities proposed clearly indicates that it has a very positive impact through the elimination or improvement of non-optimized actions. Upon questioning the workers, we found their opinions to be directed towards various categories: greater fluidity of the conveying process through the elimination of small, repetitive and tedious steps; greater interconnection between the most relevant IT processes; and the handover of decision making to the IT system in the back office. In the Lasts and Finishing area, the addition of some necessary machines and the layout change in the area has created the necessary workflow and harmony. Also using the Lean VSM, Gemba walk and standard work tools, Rahani and al-Ashraf [52] managed to reduce operator working time by 16.9% and machine time required to assemble a front disc brake system by 14.2%. In this case, a "cocktail" of Lean tools was also used to achieve the desired results in terms of decreasing the cycle time that was necessary.

The implementation of the TPM tool allowed us to register the work performed and what remains to be done, planning the tasks according to priority and the time needed for each intervention. The managers and team leaders highlighted the ease of opening labels and the feeling that their requests are no longer forgotten. These results are corroborated by other research [6,7,48] where a better organization of the maintenance service has induced higher levels of productivity due to less time being wasted on waiting by overcoming equipment breakdowns, tasks scheduling, spare-part management, and so on. This tool can also be supported by other Lean tools, such as SMED (Single Minute Exchange of Die) and others. In the case of this work, allied to the visual management tool, it has produced the required results while also making the workers' workflow easier.

The introduction of the labels on delayed pairs also used the visual management tool and this allowed for a faster identification of these late pairs and for the workers to have a greater focus on the identified work. This urgency becomes more important due to the need to enable the company meet the date agreed upon with the customer. Monteiro et al. [53] also used visual management and other Lean tools to reduce the non-conformities in a metalworking firm. The implementation of new procedures has induced a decrease of 2.04% in the non-conformities detected internally and 3.99% in the non-conformities detected by the customer, improving in this way the image of the company and its performance in the market.

Regarding the Yokoten document, Figure 8 shows the number of accidents from January 2021 to August 2022, with an average value of 1.4 accidents per month. In the period from September 2021 to July 2022, an average of 0.7 accidents per month was reported. This report signals a reduction by half regarding to the previous period. That is a 50% improvement in safety. Sá et al. [37], analyzing a furniture company, stated that 40%

of the workers considered that the implementation of the tools made the workspace more safe.

As in Morgado [46], when the companies are asked about "What are the benefits of a workplace health and safety management system implemented in your company?", some answers were also found in this study, specifically, "Reduction in work accidents", "Improved workers satisfaction" and "Increased productivity".

It is necessary to promote the identification of possible hazards and respective risks and to provide safety training to workers in order to avoid risks and problems.

*Monitoring*

Monitoring is the last step of the action research methodology. The SVSM map should be updated after fixed periods of time, whenever the people in charge decide so and in the case of regression in terms of productivity or safety indicators. A continuous improvement team should be called to the field in order to approach the situation. This group can act with the intention of intervening through the application of the Lean tools already mentioned, such as the Gemba walk, or others not yet applied and understood as beneficial for each case to be overcome.

To complement the map, which may not be updated daily, a KPI should be built to evaluate the production pace of an area according to the number of pairs made over the time elapsed, and at the safety level this way carried out, to indicate the area's risk levels.

To monitor the visual management tool, surprise internal audits should be carried out to assess whether late pairs are identified and whether workers are aware if they are working on a late pair.

The Yokoten document should be updated whenever there is a new accident and, in the kaizen meeting at the beginning of the following day, the workers should be informed of the event so that it may serve as a precaution against the occurrence of more accidents due to the same reason.

Although this research has achieved its objectives, showing how lean practices have an impact on issues of occupational safety of workers, future research could involve the study and application of SVSM in other types of companies in order to consolidate this study, and help to verify what gains the Lean philosophy brings to the safety of workers at their workplaces. Other tools are suitable for use in this in addition to the four used. Ahmad et al. [54] stated that the use of poka-yoke was able to reduce the risk score from 9 to 3, a reduction of 55.6% in the risk. Kaizen is another tool that can be used, with James [29] stating that the tool improved safety in construction-related industries.

It is important to make it clear that the impact of the different Lean tools used is different. It becomes evident that the Gemba walk and TPM tools provide the best results and offer the least limitations in terms of applicability and universality of solutions. On the other hand, the Yokoten tool presents higher limitations, as it does not produce direct effects on productivity and does not act directly on risk factors, eliminating them or drastically reducing the chances of them occurring. It can be seen that the tools used have a greater relevance to the productivity factor than to directly reducing risk. However, by acting on the productivity factor, tools such as TPM and Gemba walk are producing indirect effects on workers' safety, which gives them greater security and satisfaction at work. Therefore, some questions arise: Is it actually necessary to use the four selected tools simultaneously? Are there other Lean tools capable of producing better effects in terms of reducing accident risks? Regarding the first question, it is clear that TPM and Gemba walk are absolutely necessary, and that the application of TPM can be assisted using visual management. However, visual management was essentially more useful in safeguarding delivery deadlines for batches of shoes that, for whatever reason, were delayed in production. The Yokoten tool had greater limitations than it did a positive impact on the results obtained. As for the second question, which clearly addresses future work, it made sense to apply the 5S tool with a view to improving the conditions of each workplace. In addition, tools such as root cause analysis and 5 whys should be considered whenever an accident occurs, allowing greater

knowledge about its causes, which could help prevent the same from happening again. The application of the PDCA (plan–do–check–act) tool to resolve problems, once the cause has been identified, could also be a valuable aid in pursuing more effective security procedures. These last tools would not be applied permanently, but only when any accident arises. The application of 5S should be intensive and permanent.

The footwear industry is largely located in developing countries, where labor is not yet extremely expensive. In medium or small companies, top management does not always have the appropriate preparation to outline long-term strategies. The methodologies mentioned here can help to assist top managers who are less prepared in terms of management tools to understand the benefits they can extract from the application of these tools, as well as their ease of application. This could induce top management to create training routines for its workers in Lean tools, which could multiply the chances of new ideas and methodologies emerging that could contribute in a much more positive way to increasing the productivity and safety of companies with these specificities.

## 5. Conclusions

After carrying out this case study, it became clear that, in the first instance, workers who are present at workstations every day have a wide knowledge about equipment and daily operations, and their opinions are a good starting point for identifying what should be improved in the production process. In this aspect, the Gemba walk and the TPM are the most effective tools because they facilitate cooperation between the workers and the management team and create continuous improvement teams in a joint objective of optimizing the production process. Figure 9 shows a summary of the advantages and disadvantages of the Lean tools applied.

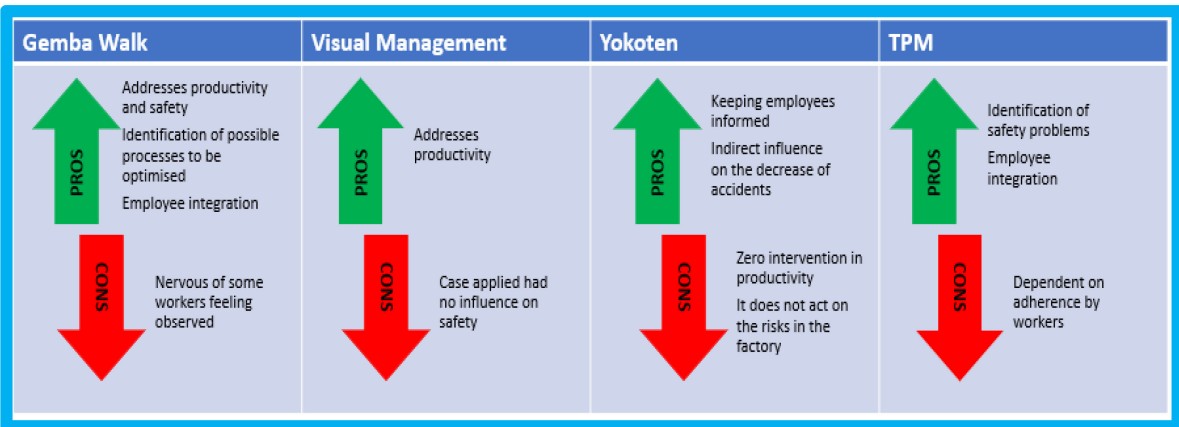

**Figure 9.** Graphical summary of advantages and limitations on the application of each Lean tool.

These four Lean tools were chosen for this case study was due to the fact that they present great adaptability and are often referred to in the literature as excellent tools for producing effective results after implementation. Nowadays, in a company that already has a Lean culture rooted or is still introducing it, the presence of a TPM system is mandatory. The Gemba walk tool is recommended for its effectiveness in identifying less optimized processes.

During and after the implementation of these tools in the present study, some limitations emerged because the sample in any two moments was not perfectly the same. The fact of being an orthopedic footwear company allows the customer to choose a high quantity of requirements for the shoes in order to make them into a unique piece. This makes it difficult to evaluate the improvements felt after the implementation of Lean tools in a statistical way because the variants to which the production process is exposed are too wide to control. Another conditioning factor is the fact that this is a handcraft company.

This kind of process is very dependent on the knowledge of each worker and the pace of work is dictated by their physical and mental availability, and not by machines.

In conclusion, this study supports the theory that organizations that implement Lean tools can achieve benefits in safety and productivity. In order to draw more conclusions about the impact of Lean tools on occupational safety in a company, more case studies should be conducted in other companies from the most varied sectors. A more adequate selection of Lean tools to the type of product manufactured and to the type of manufacturing could produce more expressive results that contribute to our conclusions as to the impact of the tools.

**Author Contributions:** The authors made the following contributions to this work: concept about the idea, J.C.S.; J.D.-C. and L.S; Data Curation, J.C.S.; G.S. and L.S.; Problem analysis J.C.S.; J.D.-C. and L.S.; Investigation J.C.S.; J.D.-C. and L.S.; Methods, J.C.S.; J.D.-C. and F.J.G.S.; Writing—original draft, J.C.S., L.S. and F.J.G.S.; Writing—proofreading improvement and editing, J.C.S.; J.D.-C. and F.J.G.S. All authors have read and agreed to the published version of the manuscript.

**Funding:** This research did not receive external funding.

**Institutional Review Board Statement:** ISEP have been contacted by the authors of the manuscript for publication at the Safety Journal. According to the Portuguese legislation, this study caried out in the field of occupational health and safety does not require a pre-approval of an Ethical Committee.

**Informed Consent Statement:** Not applicable.

**Data Availability Statement:** The data are available on request.

**Conflicts of Interest:** The authors declare no conflict of interest.

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
