# Peer review of "Assessment of the Impact of Lean Tools on the Safety of the Shoemaking Industry"

_safety, 2023_

Round 1

Reviewer 1 Report

It is proposed to publish in journal in the current version.

Author Response

Dear Editor,
The authors would like to thank the editor and reviewers for the time spent reviewing this article and for the positive feedback, helpful suggestions, and valuable criticism. We have carefully considered the comments of the reviewers and believe that the article has been completely improved.
In the attached file, you can find the questions posed by the reviewers, as well as our answers. We also present a complete revised version of the article, where we tried to include all the suggestions made by the reviewers.

Reviewer 2 Report

The paper deal with an interesting concept but has many flaws ned to adjust:

-The literature basis of the paper is very small - it should be stenghted and Authors shold include more good papers from international journals. Expand the literature review to provide a more thorough overview of existing research on Lean tools and their impact on safety and productivity in the footwear manufacturing industry. Include studies that both support and challenge the findings of the current research.

-The paper should clearly differentiate between causality and correlation. It should be cautious about making causal claims if the study design does not allow for causal inference.

-Any limitations in the methodology used in the study, such as potential biases, should be discussed transparently. These might include issues like self-reporting bias or measurement errors.

- Provide a more detailed description of the research methodology, including data collection procedures, tools used, and statistical analysis methods. This transparency can enhance the paper's rigor.

- Discuss the practical implications of the research findings. How can organizations in the industry benefit from implementing Lean tools? Provide actionable insights.

-Improve discussion section. Discuss results with more other researches. Provide a more in-depth discussion of the implementation of Lean tools, including the specific strategies employed, the challenges faced during implementation, and how these challenges were overcome.

-Suggest directions for future research. What are the next steps in exploring the impact of Lean tools on safety and productivity?

- Dedicate a section to discussing the practical implications of the research for managers and decision-makers in the footwear manufacturing industry. Offer actionable insights on how companies can apply the lessons learned from the study.

It's ok.

Author Response

Dear Editor,
The authors would like to thank the editor and reviewers for taking the time to review this article and provide positive feedback, helpful suggestions and valuable criticism. We have carefully considered the reviewers' comments and believe that the article has been thoroughly improved.
In the attached file you can find the questions posed by the reviewers, as well as our responses. We also present a complete revised version of the article, where we have tried to include all the suggestions made by the reviewers.

Reviewer 3 Report

The manuscript is interesting and usable. However, there are a few details that I would like to draw your attention to:

- The limitations of the research are not stated, neither in the abstract nor later, and I think they can guide future research.

- The implications of the research are not stated, because they can lead to future direct application and diffusion of knowledge.

- The literature at the end of the work is arranged according to a style unknown to me. If it's a style that this magazine tolerates, then fine, and if it's not, please fix it.

- I would also like to draw attention to the fact that the list of literature could have been wider than thirty references to literature.

- Figures 1 and 7 are in Spanish, which is not a problem, if the magazine tolerates two languages appearing in the text. If not, please translate the figures into English.

- The initial hypothesis must be particularly emphasized in the methodological setting, not in the results. The hypothesis is confirmed or rejected in the discussion.

I came across several sentences that were not clear to me. I ask that the review and editing of the text be done once more.

Author Response

(The authors gave the same response as above.)

Reviewer 4 Report

This study applies Lean tools and analyzes their impact on the productivity and safety of an orthopedic footwear company.

In the abstract it should be stated the background of the problem. Then in the introduction it should be better shown what is said in the abstract (and from there it should be narrowed down) regarding the instrument developed in the previous studies, to clarify where to start.

The action research methodology is described in the stages of the quality cycle, which should be better specified.

Also in paragraph 2.1.1. it can be shown that Lean is also used in combination with Six Sigma for improvement of medical organization and assessment of labor practices in healthcare with the support of frameworks for sustainability, as revealed in the paper: Moldovan F, Moldovan L, Bataga T. Assessment of Labor Practices in Healthcare Using an Innovatory Framework for Sustainability. Medicina (Kaunas). 2023 Apr 19;59(4):796.

Consecutive paragraphs 2.3. Action Planning and 2.4. Action Planning have the same title.  The last one should be different according to the methodology.

The results are clearly presented, but after the title 3.Results there is only one subsection 3.1. Results presentation that makes no sense and should be deleted as a title.

There is also a discussion paragraph, which also has a single subsection 4.1. Monitoring, which should be followed by another subparagraph. Also from this perspective, the paper should be rearranged.

The "Limitations of This Work and Future Research" section could be included at the end of the discussion paragraph without being a separate paragraph.

In figures 1 and 7 fields should be translated in English.

In fig 8 – Fev should be in English.

Other editing errors:

Line 79: on the shop floor., [15].

Line 89: indicators[21].

Line 134: Work)..

In references some dates should be translated in English also, ex.: J Clean Prod. 10 de Outubro de 2021; Disponível em:; Pesquisa-ação na engenharia de produção: proposta de estruturação para sua, etc.

Author Response

(The authors gave the same response as above.)

Round 2

Reviewer 2 Report

Authors made good job to improve paper according my remarks.

It's ok.

Reviewer 3 Report

Dear authors,

thank you for accepting my suggestions.

My deepest apologies for writing Spanish instead of Portuguese by mistake - thanks for editing the pictures...

All the best.

Reviewer 4 Report

The authors have responded to the recommendations for improving the paper I have carried out in round 1.